# The Effect of Sensory Satiety on Perceived Benefits: The Case of Aesthetic Consumption in South Korea

**Joo-Eon Jeon** [1],* and **Eun Mi Lee** [2]

[1]   Department of Global Business Administration, Anyang University, Anyang 14028, Korea
[2]   International College, Dongseo University, Busan 47011, Korea; eilly0328@gmail.com
*   Correspondence: eric@anyang.ac.kr

**Abstract:** Repeated exposure to aesthetic design results in consumers experiencing satiation because of sensory satiety. In other words, being consistently exposed to aesthetic stimuli activates consumers' sensory satiety, defined as the drop in sensory pleasure, and the resulting reduction of their value of aesthetic products ultimately leads to switching intentions. That is, sensory satiety reduces functional and emotional benefits. Furthermore, consumers are unlikely to recall every item they have consumed, and are instead likely to focus on a particular option. Thus, this study predicts that consumers can recover from satiation over time. This research proposes that both satiation and accustomedness negatively affect functional benefit. As an empirical study, the research uses a multiple regression model for two purposes: The first is to test the impact of sensory satiety on perceived benefits, and the second is to observe the change in sensory satiety over time. We find that satiation and accustomedness, as sub-dimensional scales of sensory satiety, reduce perceived benefits. The results showed that it is clear that only satiation reduced functional benefits, whereas both satiation and accustomedness reduced emotional benefits. In addition, our study confirms the change in sensory satiety over time. Consumers who have been continuously exposed to, and used, aesthetic products become accustomed to them and feel satiated. Based on these results, this study will be useful for the sustainability of the product life cycle.

**Keywords:** accustomedness; perceived benefits; satiation; sensory satiety

## 1. Introduction

Due to a rapid change in customers' needs and the environment, many companies have recently made various efforts to propose new customer values and build strong customer relationships. Companies are increasingly interested in exceptional marketing programs, which are designed to provide the customers with values that can induce mutual understanding beyond simply purchasing and using products [1,2]. While building successful experiential marketing that will help establish customer loyalty, companies are also becoming interested in developing aesthetic benefits that will satisfy consumers' sensory pleasure needs [3–6].

The daily life of the average consumer is filled with aesthetic design. These aesthetic products typically aim to increase product awareness and purchase intentions, often by highlighting hedonic consumption and inviting consumers to imagine the sensory experience for themselves. Aesthetic design, which can satisfy consumers' aesthetic needs, has been favorably evaluated because it can quickly match a brand to its target audience amid various marketing incentives exposed indiscriminately across the market [7]. Previous research has verified the effectiveness of aesthetic design in current trends [3–11].

Existing studies show that aesthetic design not only allows target customers to recognize the brand (product) positively, but also builds strong customer relationships through favorable images. However, repeated exposure to aesthetic design results in consumers experiencing satiation because of sensory

satiety [12–15]. In other words, consumers experience satiety, defined as the drop in sensory pleasure, with repeated aesthetic consumption [16]. Krishna [4], who attempted various approaches to study consumers' aesthetic experience, insisted on the necessity of satiation research due to sensory satiety with regard to aesthetic marketing. However, in the business field, there is a lack of consideration and theory about sensory satiety and satiation [17]. Sensory satiety is largely divided into a physiological response and a cognitive response. Most of the previous studies have focused on physiological responses, and few have been interested in sensory satiety through cognitive responses. Based on the aforementioned studies, Lee and Jeon [18] used the concept of sensory satiety as the satiation that results from being accustomed to sensory pleasure provided by aesthetic products and feeling sensory fatigue from repeated use to develop the two concepts of composition: accustomedness and satiation.

A consumer who has experienced sensory satiety in a product has a willingness to explore other products of their own accord. This leads to customer defection and makes it difficult to build customer loyalty. Therefore, companies that recognize the importance of aesthetic products will need to manage consumer sensory satisfaction appropriately. In this study, the sensory satiety experienced by consumers is focused on the possibility of developing amnesia over time. Consumers who have repeatedly encountered aesthetic products store this information in their working memory and experience satiation [16,19,20]. However, these memories may be subjected to amnesia due to interference in the process of memory retrieval through decay and new afferent information. Despite experiencing satiety in aesthetic products, the amnesia that consumers can develop over time can lead to renewed interest. Therefore, unlike earlier studies, this paper aims to verify the recovery from sensory satiety in aesthetic products.

Our research asks a simple but fundamental question: Does sensory satiety of aesthetic products reduce perceived benefit? Does sensory satiation change over time because of recovery from sensory satiety? Thus, this study investigates whether sensory satiety reduces perceived benefits. In addition, the authors predict that the impact of satiation and accustomedness on perceived benefits will vary over time. For empirical testing, the authors use a multiple regression model to examine the hypothesis. Multiple regression is used to both test the impact of sensory satiety on perceived benefits and observe the change in sensory satiety over time. This study will be useful for the sustainability of the product life cycle.

## 2. Background Theory

### 2.1. Sensory Satiety of Aesthetic Design

Many companies that have reached the limit of differentiating based on product quality are beginning to focus on aesthetic design, as consumers respond favorably to synesthesia from afferent marketing stimuli. To exemplify this, although during a point-of-sale transaction, most consumers typically rely on visual input to generate first impressions of an aesthetic product, repeated input can provide confirmation of the initial visual impression [21], thereby creating sensory satiety.

Spence [22] suggested that managers can enhance their consumers' product experiences by ensuring that the sound symbolism of the brand name, as well as any shape symbolism of the labeling and the shape of the packaging itself, forms the congruent product-related sensory expectations in the customer's mind. The sensory organ undergoes a perceptual process of interpreting the stimuli introduced in the process of detecting and transmitting marketing stimuli to the brain in a biochemical reaction, which leads to a favorable response if the stimuli are perceived to be in harmony [4]. Ultimately, if the sensory factors are harmonized in the process of the marketing stimuli becoming an effect, then the sensory factors will result in the efficiency of the sensor [1,23]. Therefore, sensory factors contribute to customer satisfaction and product acceptance [24,25]. The aesthetic experience builds a strong and favorable image, creating a strong customer relationship. An aesthetic response that has never been experienced not only incites curiosity in the consumer, but also leads to the purchase of the product after the discovery [26–28].

However, frequent exposure to the same aesthetic stimulus may lead to consumers searching for new stimuli to dull their first pleasure [19,20]. Therefore, repeated exposure to aesthetic design results in consumers experiencing satiation caused by sensory overload. Satiation reflects a cognitive process of assessing how many aesthetic products have been consumed in the past [16]. Thus, aesthetic stimuli can satisfy an individual's optimal stimulation level, leading to the rejection of the product [29,30]. The attentional capture of aesthetic design is modulated by sensory satiety [31].

According to previous studies, sensory satiety is divided into a physiological response and a cognitive response. Most of them focused only on physiological responses, and there is relatively little discussion about sensory satiety through cognitive response [13,14,32]. There have been attempts to measure the sensitivity of cognitive reactions, but only to explain whether sensory satiety is actually caused by cognitive response by measuring the intake of specific foods [33]. Civille and Oftedal [34] suggested sensory evaluation techniques that can measure sensory responses through cognitive processes.

Jeon and Kim [35] developed a satiety quotient that measures consumers' product experiences. They proposed a satiation measure by indexing the satiation period compared to the product ownership period expected by the consumer. However, it is difficult to explain the sensory satiety of the aesthetic product because they focused on the period of product possession without considering the response to the aesthetic design.

Jeon and Lee [17] identified satiety in terms of cognitive overload, observing fatigue from the aesthetic stimuli of repeated exposure and the degree of behavior required to avoid sensory satiety. They also argued for the need to develop measurement questions that can be measured quantitatively, believing that repeated exposure to the aesthetic design could lead to cognitive overload. In this study, the sensory satiety experienced by the consumer is defined as "satiation that occurs due to the accustomed experience of sensory pleasure due to repeated use of aesthetic products and the feeling of sensory fatigue." Exploratory studies have found that the satiety of aesthetic design consists of accustomedness, sensory fatigue, and satiation. A detailed description of each construction concept is as follows.

First, sensory satiety is accustomedness. The accustomedness is defined as a situation in which consumers who have repeated exposure to the aesthetic design no longer anticipate new and enjoyable experiences. Consumers with prolonged exposure to aesthetic design do not respond to sensory pleasure. Second, consumers who are accustomed to aesthetic design and experience sensory fatigue might be in the stage of full-scale deterioration of their sensory response. The level of sensory pleasure in the aesthetic design experienced in the early stage decreases. Third, with satiation, the sensory response to the aesthetic design becomes boring and is no longer interesting. These three constructs of sensory satiety explain the avoidance of aesthetic design. The development of satiety constructs has contributed to the expansion of previous research on sensory satiety in the area of cognitive response along with the emotional response. It was also verified that satiety not only reduced sensory pleasure, but also confirmed that the switching intention could be formed by minimizing the perceived practical and emotional value.

Functional benefit should emphasize functional performance. Holbrook and Hirschman [1] defined functional value as the ability to perform functions in the everyday life of a consumer. Functional needs and basic motivations are linked and are then met with products and functional performance [36,37]. Emotional benefit was defined as a steady flow of fantasies, feelings, and fun [1]. This emotional benefit regards consumption as a primarily subjective state of consciousness with hedonic responses and aesthetic criteria.

Consumers exposed to aesthetic design initially respond positively to sensory pleasure. However, consumers who are consistently exposed to aesthetic stimuli experience sensory satiety, and this reduction of customer value in aesthetic products ultimately leads to switching intentions. That is, sensory satiety reduces functional and emotional benefits. Furthermore, this study predicts that

consumers who experience satiety of an aesthetic design will be less likely to perceive benefits. Based on these findings, this study identifies the amnesia effect of sensory satiety.

**Hypothesis 1 (H1):** *Sensory satiety, including satiation and accustomedness, will reduce functional benefits.*

**Hypothesis 2 (H2):** *Sensory satiety, including satiation and accustomedness, will reduce emotional benefits.*

*2.2. Variety of Sensory Satiation over Time*

The satiation caused by consumption of a product with an aesthetic design leads to sensory satiety, and the sensory satiety formed from cognitive overload may lead to its memory being diluted after a certain time due to amnesia [19,20]. If satiation depends on recalling past consumption, then it is appropriate to ask what consumers will spontaneously recall. Time takes the future consequences of a particular choice into account [38]. In this respect, it seems that time is related to the customer's willingness to delay or expedite gains (such as obtaining a reward or something of value) and losses (giving up something of value) [39]. Consumers are unlikely to recall every item they have consumed, and are instead likely to focus on a particular option. Thus, the authors predict that consumers can recover from satiation over time.

Recovery from sensory satiety is based on a working memory mechanism [29]. Memory is the process of storing and retrieving information based on experiences, and usually involves two stages, from learning to inspection to retrieval. Only some episodic memory, which is actuated through the storage organ, is stored as working memory, and the withdrawn episodic memory facilitates recall. In the process of consumption, there is also storage and withdrawal by these working memories in which the individual's various spending experiences are encoded, and selective attention is paid to the working memory [16]. Sensory satiety as the repeated use of the aesthetic design is also stored in the consumer's working memory. Interestingly, the speed of experiencing sensory satiety varies depending on the individual's working memory ability. In other words, individuals with high working memory abilities absorb the external stimuli faster than individuals with low working memory abilities because of their superior afferent, learning, and withdrawal capabilities [29,39].

Existing studies have shown that the sensitivity of an individual's experience is affected by amnesia over time. That is, time can accelerate recovery from satiation. Consumers buy and store a variety of products (services) in working memory in anecdotal form during their experience, and it is very difficult to withdraw all of these experiences from memory. In particular, memory withdrawal of the consumption experience is harder when it is not linked to a special and significant clue than otherwise [29,40]. Thus, recovery from sensory satiety experienced by consumers can be described as a memory withdrawal failure. That is, if one does not find the information encoded in their memory and the withdrawal cues for their satiety, they experience amnesia. Therefore, sensory amnesia might increase over time.

Recovery over time is explained in two ways. The first is the decay of memory, and the second is interference. The detailed description is as follows. First, recovery by decay is when the memory of external information changes the central nervous system through the sensory organ, leaving a memory trace, but fades over time and eventually disappears through the metabolic process [41]. Based on these discussions, repetitive exposure to the aesthetic design creates memory for sensory satiety, but, over time, decay can lead to difficulty in withdrawing sensory satiety. Second, the recovery of sensory satiety generally comes from the interference effect because the previously learned information is interfered with by information on newly perceived sensory satiety. The difficulty of withdrawing information of the past consumption experience can thus be described as interference [40]. Sensory satiety related to aesthetic products can also be explained in terms of interference effects. Consumers who purchase aesthetic products repeatedly and are frequently exposed might experience sensory satiety from the product. The memory of the sensory satiety, however, is interfered with as various

consumption experiences directly or indirectly related to the aesthetic products are accumulated, which is necessary to retrieve the memorized past sense, thereby generating amnesia.

In summary, recovery from sensory satiety can be defined as a situation in which the memory of the sensory satiety stored by working memory fades, resulting in the consumer experiencing withdrawal difficulties. Due to various interferences, consumers who experience less of an aesthetic product become amnesic of the clues of the sensory satiety that have been in their memory over time for that product. Given these points, the following research hypothesis is provided.

**Hypothesis 3 (H3):** *The impact of satiation and accustomedness on perceived benefits varies over time.*

## 3. Materials and Methods

### 3.1. Data Collection

The purpose of this study is to verify that, although repeated exposure to aesthetic design diminishes consumers' interests, a certain level of amnesia reduces the degree to which consumers' interests diminish. In this study, respondents who purchased aesthetic products were selected as samples. The period of purchase was used as a measure to identify the difference in sensory satiety over time. To be specific, this research sought to classify consumers who had purchased an aesthetic product within the last month, within the last 1–3 months, and within the last 3–6 months. The survey was conducted using a marketing research institution known as a panel company. Researchers collected data from panel members who agreed to take part in surveys for compensation. The panel members are motivated by points that accumulate as they take part in surveys. Based on a study by Lee and Jeon [18], the respondents were consumers who purchased Apple AirPods, Dyson vacuum cleaners, or Starbucks tumblers within the last six months—the main reason for selecting these is that many customers consider them as aesthetic products. A total of 298 respondents participated in this online questionnaire. The group of respondents consisted of 110 males (38.1%) and 179 females (61.9%). In terms of the age ranges, 113 (39.1%) were 25–29 years old, 100 (34.6%) were 30–34 years old, and 76 (26.3%) were 35–39 years old.

### 3.2. Procedures and Variables

The questionnaire items were developed in English. Because this study was conducted in South Korea, a translation–back-translation process was used [42]. First, all the questionnaire items of the survey were translated into Korean, and were later translated back into English by two bilingual researchers to cross-check the accuracy of the translations. Data were collected from panel members who had registered with the research institute, with their consent. The questionnaire was conducted with the cooperation of participants. The authors explained the purpose of the research to the participants and informed them that they could withdraw their participation at any time, that all personal data would be kept confidential according to the Korean Statistical Law, and that all data would be destroyed after one year. There was a screening test for participants who expressed their interest to join to select a suitable panel for this study. Before answering the questionnaire, participants were asked to consider the aesthetic products they already owned. After doing so, each question on sensory satiety, functional benefit, and emotional benefit of the products selected by the participants was measured. When all the survey responses were completed, the marketing research institute expressed their gratitude and provided compensation for participating in the survey.

The measurement items of the main constructs used in this study are as follows:

*Sensory satiety* is defined as accustomedness, and satiation is the situation in which a consumer experiences repeated exposure to aesthetic stimuli, based on measurement questions developed by Jeon and Lee [17]. These measurements distinguished between several sensory satiety dimensions and led to the construction of a sensory satiety scale. Jeon and Lee [17] showed that the scale is reliable, valid, and distinct from other constructs in South Korea. Accustomedness refers to no longer

expecting sophisticated sensory stimuli, and was measured by two items: "This product does not feel unique" and "This product is no longer new." Satiation is then defined as finding the aesthetic stimulus uninteresting, and was measured by four items: "I am tired of the design of this product now," "This product no longer looks sophisticated," "I am not currently using this product for more fun," and "This product is no longer attractive."

*Perceived benefit* is defined as the type of convenience consumers perceive in a product [36,43]. Functional benefit is defined as the convenience perceived by functional attributes, such as product function and quality, and emotional benefit refers to the emotional convenience expected in the process of product consumption [1]. For the functional benefit, three items were selected: "This product is useful," "This product is functional," and "I may rely on this product's function"; for the emotional benefit, three items were also selected: "This product makes me happy," "This product is attractive to me compared to other products," and "This product gives me a special feeling" [36]. All questions consisted of a five-point Likert scale.

### 3.3. Analysis

The data set consisted of individuals. We used a multiple regression model to examine the hypotheses. Multiple regression was used for two purposes: The first was to test the impact of sensory satiety on perceived benefits, and the second was to observe the change in sensory satiety over time.

The analysis included demographic characteristics, such as sex and age, as control variables (all categorical variables were converted into dummy variables). To test the hypotheses, we regressed each consequence variable on the predictor variables; all predictors were group-mean centered (individual mean).

## 4. Results

### 4.1. Analysis of Reliability and Validity

We evaluated reliability and validity based on two criteria: (1) Composite reliability and Cronbach's alpha (α) should be greater than 0.7 in reliability testing (satiation should be greater than 0.898, accustomedness should be greater than 0.791, functional benefit should be greater than 0.820, and emotional benefit should be greater than 0.862); (2) all factor loadings should exceed 0.4. Table 1 shows the results of factor analysis between accustomedness and satiation, which represent the factor of sensory satiety. They were analyzed via principal component analysis and maximum likelihood extraction methods, followed by Varimax rotation. For the factor analysis, the accustomedness and satiation that consist of the sensory satiety were extracted normally, and each factor accounts for 78.80% of the variance. The values of the means and standard deviations are shown in Table 2. Based on the results, it is obvious that the concept of sensory satiety could be considered as a multidimensional scale with accustomedness and satiation.

**Table 1.** Results of the factor analysis.

| Items | Satiation | Accustomedness |
|---|---|---|
| I am tired of the design of this product now. | 0.682 | 0.407 |
| This product no longer looks sophisticated. | 0.834 | 0.342 |
| I am not currently using this product for more fun. | 0.859 | 0.306 |
| This product is no longer attractive. | 0.818 | 0.370 |
| This product does not feel unique. | 0.357 | 0.824 |
| This product is no longer new. | 0.351 | 0.847 |
| % | 46.999 | 31.801 |

**Table 2.** Descriptive results of factors.

| Factors | Time | *n* | Mean | S.D. | S.E |
|---|---|---|---|---|---|
| Satiation | 1 month | 93 | 2.11 | 0.81 | 0.08 |
| | 1–3 months | 97 | 2.40 | 0.82 | 0.08 |
| | 3–6 months | 99 | 2.22 | 0.80 | 0.08 |
| | Total | 289 | 2.25 | 0.82 | 0.04 |
| Accustomedness | 1 month | 93 | 2.50 | 0.88 | 0.09 |
| | 1–3 months | 97 | 2.92 | 0.96 | 0.09 |
| | 3–6 months | 99 | 2.76 | 0.93 | 0.09 |
| | Total | 289 | 2.73 | 0.94 | 0.05 |
| Functional Benefit | 1 month | 93 | 3.67 | 0.78 | 0.08 |
| | 1–3 months | 97 | 3.52 | 0.72 | 0.07 |
| | 3–6 months | 99 | 3.62 | 0.73 | 0.07 |
| | Total | 289 | 3.60 | 0.74 | 0.04 |
| Emotional Benefit | 1 month | 93 | 3.44 | 0.84 | 0.08 |
| | 1–3 months | 97 | 3.29 | 0.70 | 0.07 |
| | 3–6 months | 99 | 3.26 | 0.77 | 0.07 |
| | Total | 289 | 3.33 | 0.77 | 0.04 |

As shown in Table 3, the results of the correlation analysis between the dependent and independent variables show that satiation and accustomedness have a significant positive correlation. In addition, it is obvious that there are significant negative correlations between satiation and perceived benefits (functional benefit and emotional benefit), and between accustomedness and perceived benefits (functional benefit and emotional benefit). Finally, functional and emotional benefits have a positive correlation. The above results indicate that sensory satiety (satiation and accustomedness) and perceived benefits (functional benefit and emotional benefit) are interrelated.

**Table 3.** Results of discriminant validity.

| | Satiation | Accustomedness | Functional Benefit | Emotional Benefit |
|---|---|---|---|---|
| **Satiation** | 1 | | | |
| **Accustomedness** | 0.725 ** | 1 | | |
| **Functional Benefit** | −0.383 ** | −0.317 ** | | |
| **Emotional Benefit** | −0.453 ** | −0.435 ** | 0.479 ** | 1 |

** $p < 0.001$.

## 4.2. Hypothesis Testing

### 4.2.1. Main Effects of Sensory Satiety on Perceived Benefits

This research proposes that both satiation and accustomedness would negatively affect functional benefit. The effect of satiation with aesthetic products on functional benefit is significant (b = −0.319, $p < 0.001$), whereas the effect of accustomedness is not significant (b = −0.086, $p = 0.283$). That is, satiation with aesthetic products seems to be a stronger predictor of functional benefit than accustomedness. Functional benefit serves as a basic and fundamental descriptor of the product category. Thus, Hypothesis 1 was not supported.

Additionally, the authors predicted that both satiation and accustomedness would be negatively related to emotional benefit. The path between satiation and emotional benefit is significant (b = −0.293, $p < 0.001$), as is the path between accustomedness and emotional benefit (b = −0.222, $p < 0.001$). Thus, Hypothesis 2 was supported.

Therefore, this result reveals that consumers who experience satiation with aesthetic products are less likely to consume for sensory pleasure. The overall results of this study demonstrate that sensory satiety with aesthetic products is negatively related to perceived benefit.

### 4.2.2. The Change in Sensory Satiety over Time

In order to verify the research purpose of this study, we would like to examine the amount of changes in the satiation and accustomedness over time. Table 4 shows the change in the impact of satiation on perceived benefits over time.

**Table 4.** Results of multiple regression.

| | Total | | | | |
|---|---|---|---|---|---|
| | Unstandardized Coefficients | | Standardized Coefficients | *t* | Sig. |
| | B | Std. Error | Beta | | |
| (Constant) | 4.843 | 0.247 | | 19.635 | 0.000 |
| Sex | −0.026 | 0.086 | −0.017 | −0.302 | 0.763 |
| Age | −0.122 | 0.051 | −0.131 | −2.394 | 0.017 |
| Satiation | −.0291 | 0.073 | −0.319 | −3.994 | 0.000 |
| Accustomedness | −0.069 | 0.064 | −0.086 | −1.075 | 0.283 |
| | F = 14.177 (*p* = 0.000), R = 0.408, R2 = 0.166, Adjusted R Square = 0.155 | | | | |
| | Dependent Variable: Functional Benefits | | | | |
| | Unstandardized Coefficients | | Standardized Coefficients | *t* | Sig. |
| | B | Std. Error | Beta | | |
| (Constant) | 4.513 | 0.246 | | 18.384 | 0.000 |
| Sex | −0.014 | 0.085 | −0.009 | −0.165 | 0.869 |
| Age | −0.014 | −0.051 | −0.014 | −0.269 | 0.788 |
| Satiation | −0.276 | 0.072 | −0.293 | −3.809 | 0.000 |
| Accustomedness | −0.182 | 0.064 | −0.222 | −2.871 | 0.004 |
| | F = 21.126 (*p* =0.000), R = 0.479, R2 = 0.229, Adjusted R Square = 0.218 | | | | |
| | Dependent Variable: Emotional Benefits | | | | |
| | Within 1 month | | | | |
| | Unstandardized Coefficients | | Standardized Coefficients | *t* | Sig. |
| | B | Std. Error | Beta | | |
| (Constant) | 4.798 | 0.399 | | 12.032 | 0.000 |
| Sex | 0.130 | 0.154 | 0.083 | 0.848 | 0.399 |
| Age | −0.110 | 0.093 | −0.112 | −1.186 | 0.239 |
| Satiation | −0.052 | 0.153 | −0.054 | −0.338 | 0.736 |
| Accustomedness | −0.363 | 0.144 | −0.409 | −2.528 | 0.013 |
| | F = 6.311 (*p* =0.000), R = 0.472, R2 = 0.223, Adjusted R Square = 0.188 | | | | |
| | Dependent Variable: Functional Benefits | | | | |

**Table 4.** *Cont.*

| Total | | | | | |
|---|---|---|---|---|---|
| | Unstandardized Coefficients | | Standardized Coefficients | *t* | Sig. |
| | B | Std. Error | Beta | | |
| (Constant) | 4.473 | 0.415 | | 10.781 | 0.000 |
| Sex | 0.156 | 0.160 | 0.093 | 0.974 | 0.333 |
| Age | −0.002 | 0.097 | −0.002 | −0.020 | 0.984 |
| Satiation | −0.118 | 0.160 | −0.114 | −0.736 | 0.463 |
| Accustomedness | −0.405 | 0.149 | −0.425 | −2.711 | 0.008 |
| | F = 8.125 (*p* = 0.000), R = 0.519, R2 = 0.270, Adjusted R Square = 0.237 | | | | |
| | Dependent Variable: Emotional Benefits | | | | |
| **1–3 months** | | | | | |
| | Unstandardized Coefficients | | Standardized Coefficients | *t* | Sig. |
| | B | Std. Error | Beta | | |
| (Constant) | 5.012 | 0.453 | | 11.070 | 0.000 |
| Sex | −0.215 | 0.154 | −0.137 | −1.394 | 0.167 |
| Age | −0.138 | 0.086 | −0.153 | −1.605 | 0.112 |
| Satiation | −0.418 | 0.117 | −0.475 | −3.567 | 0.001 |
| Accustomedness | 0.097 | 0.099 | 0.129 | 0.980 | 0.330 |
| | F = 4.876 (*p* = 0.001), R = 0.418, R2 = 0.175, Adjusted R Square = 0.139 | | | | |
| | Dependent Variable: Functional Benefits | | | | |
| | Unstandardized Coefficients | | Standardized Coefficients | *t* | Sig. |
| | B | Std. Error | Beta | | |
| (Constant) | 3.973 | 0.439 | | 9.059 | 0.000 |
| Sex | 0.104 | 0.149 | 0.068 | 0.698 | 0.487 |
| Age | 0.036 | 0.083 | 0.041 | 0.434 | 0.666 |
| Satiation | −0.213 | 0.114 | −0.247 | −1.873 | 0.064 |
| Accustomedness | −0.154 | 0.096 | −0.209 | −1.605 | 0.112 |
| | F = 5.313 (*p* =0.001), R = 0.433, R2 = 0.188, Adjusted R Square = 0.152 | | | | |
| | Dependent Variable: Emotional Benefits | | | | |
| **3–6 months** | | | | | |
| | Unstandardized Coefficients | | Standardized Coefficients | *t* | Sig. |
| | B | Std. Error | Beta | | |
| (Constant) | 4.698 | 0.482 | | 9.744 | 0.000 |
| Sex | 0.046 | 0.153 | 0.030 | 0.300 | 0.765 |
| Age | −1.00 | 0.092 | −0.108 | −1.078 | 0.284 |
| Satiation | −0.287 | 0.121 | −0.313 | −2.369 | 0.020 |
| Accustomedness | −0.075 | 0.105 | −0.096 | −0.721 | 0.472 |
| | F = 4.469 (*p* =0.002), R = 0.400, R2 = 0.160, Adjusted R Square = 0.124 | | | | |
| | Dependent Variable: Functional Benefits | | | | |

**Table 4.** *Cont.*

| | Total | | | | |
|---|---|---|---|---|---|
| | Unstandardized Coefficients | | Standardized Coefficients | *t* | Sig. |
| | B | Std. Error | Beta | | |
| (Constant) | 5.048 | 0.468 | | 10.796 | 0.000 |
| Sex | −0.245 | 0.148 | −0.154 | −1.654 | 0.101 |
| Age | −0.081 | 0.090 | −0.085 | −0.909 | 0.366 |
| Satiation | −0.384 | 0.118 | −0.401 | −3.268 | 0.002 |
| Accustomedness | −0.103 | 0.101 | −0.125 | −1.014 | 0.313 |
| F = 8.836 (*p* = 0.000), R = 0.523, R2 = 0.273, Adjusted R Square = 0.242 | | | | | |
| Dependent Variable: Emotional Benefits | | | | | |

Within one month, the effect of satiation with aesthetic products on functional benefit is not significant (b = −0.054, *p* = 0.736), whereas the effect of accustomedness is significant (b = −0.409, *p* < 0.05). Next, the effect of satiation with aesthetic products on emotional benefit is not significant (b = −0.114, *p* = 0.463), whereas the effect of accustomedness is significant (b = −0.425, *p* < 0.05). The results reveal that the accustomedness to aesthetic products seems to be a stronger predictor of perceived benefit than satiation.

Between one and three months, the effect of satiation with aesthetic products on functional benefit is significant (b = −0.475, *p* < 0.005), whereas the effect of accustomedness is not significant (b = 0.129, *p* = 0.330). Next, the effect of satiation with aesthetic products on emotional benefit is not significant (b = −0.247, *p* = 0.064). In addition, the effect of accustomedness is not significant (b = −0.209, *p* = 0.112). The results reveal that sensory satiety with aesthetic products seems to diminish between one and three months.

Between three and six months, the effect of satiation with aesthetic products on functional benefit is significant (b = −0.313, *p* < 0.05), whereas the effect of accustomedness is not significant (b = −0.096, *p* = 0.472). Next, the effect of satiation with aesthetic products on emotional benefit is significant (b = −0.401, *p* < 0.05). However, the effect of accustomedness is not significant (b = −0.125, *p* = 0.313). The results reveal that satiation with aesthetic products seems to be a stronger predictor of perceived benefit than accustomedness between three and six months.

The authors defined satiation and accustomedness in the sub-dimension of sensory satiety—the upper level—and analyzed the difference in sensory satiety over time. As with the results of the difference in satiation and accustomedness, the change in sensory satiety is significant over time. The results show that the impact of accustomedness within one month was significantly higher than that within 1–6 months. However, the impact of satiation was significantly higher than that over one month.

## 5. Discussion

### 5.1. Results

Until relatively recently, because they recognized that there were limitations in product-quality-based differentiation, many companies have focused on aesthetic designs that provide sensory benefits in order to gain competitive advantages that cannot be imitated by competitors. The field of business and marketing, which jumped on these industrial atmospheres, has been interested in demonstrating the effect of aesthetic design. Existing research is in a unique position to investigate how aesthetic design has had a great impact on positive performance [1,6,7,11]. Krishna [4] claimed, however, that repetitive exposure to aesthetic design might stimulate consumers' optimal stimulation

level, thereby leading to sensory satiety. In addition, Jeon and Lee [17] developed a sensory satiety scale that can be conceptualized and measured.

In this study, we expected it to be possible to show that sensory satiety from aesthetic design can be distinguished between physiological and cognitive responses, and cognitive satiety could be forgotten due to interference in working memory. Therefore, this study sheds light on the sensory satiety amnesia of those who have purchased and consumed aesthetic products and the changes in sensory satiety over time, as well as the perceived benefits (functional and emotional benefits).

This study has several key findings. First, it was confirmed that satiation and accustomedness, as sub-dimensional scales of sensory satiety, reduced perceived benefits. For this survey, we divided the participants, who possessed aesthetically designed products, into three groups (one month, 1–3 months, 3–6 months), and asked them to answer the questions on sensory satiety and perceived benefits. The results showed that it is clear that only satiation reduced functional benefit, whereas both satiation and accustomedness reduced emotional benefit. These two types of benefits were employed because consumers are aware of these benefits, thus representing the subjective compensation and expectations that one would expect before purchasing a product. Consumers would want to pursue specific compensation or expectations that can meet their particular needs (functional or emotional), so they are willing to pay attention to information searching of the product or brand that they prefer. It is unlikely that perceived benefits change after purchase, regardless of whether they meet the consumers' desires, as the initial benefits derived from the product are not affected by time.

Second, we found a change in sensory satiety over time. The results showed that the impact of accustomedness within one month was significantly higher than that found within 1–6 months. However, the impact of satiation was significantly higher than that over month. The impact of accustomedness increases within 1–3 months, but is significantly reduced just after three months. Given the points mentioned above, consumers who have been continuously exposed to—and used—the aesthetic products become accustomed to their products and feel satiated. However, the impact of satiation increases one month later.

Third, satiety stored in episodic memory might gradually diminish with decay of and interference with recalling the experience of the new product. Although there may be a difference in the time taken to experience sensory satiety, depending on the individuals' ability of working memory, this study found that the accustomedness gradually decreased three months later.

Fourth, there was no significant difference in the perceived benefits of aesthetic products, despite the effects of time, when perceived benefits were divided into functional and emotional benefits.

## 5.2. Implications

This study contributes to several academic fields. First, it was possible that sensory satiety was not a physiological response, but a cognitive response that decreased over time. In prior literature, sensory satiety from aesthetic design has been widely used to examine physiological views instead of cognitive views [18,27,36]. Although many researchers equate satiation with reaching a physiological limit, some evidence suggests that satiation can also result from cognitive processes. If consumers' sensory systems are repeatedly exposed to a product's or brand's aesthetic design, satiation from accustomedness will follow due to the activation process of various pieces of information related to the products. According to Galak et al. [29], sensory satiety can be reduced again because, as a cognitive response, consumers can lose information related to the stimulus. This is meaningful for empirically revealing that sensory satiety from aesthetic products may change over time.

Second, it has been shown that the differences in perceived benefits are affected by sensory satiety. There are various benefits that consumers expect in advance. In this study, sensory satiety dealt with accustomedness and satiation resulting from repeated use of aesthetic products. However, perceived benefits refer to subjective compensation in customers' expectations that the product will satisfy their needs [4,10,18]. Thus, our theory provides some insight into the effects of sensory satiety on perceived benefits; furthermore, it offers prescriptive suggestions. Breaking off from a repeated activity can

reduce hedonic adaptation, but consumers do not appreciate this [44]. However, our focus is on the fact that the aesthetic design of products should be tailored to the consumers' working memory ability. Additionally, sensory satiety can lead to dynamic changes (e.g., decreases in pleasure and interest) over time due to continuous use of a product, while the difference in perceived benefits is static regardless of time [16,19,29]. This research provides insights for understanding how consumers recover from aesthetic design satiation.

In terms of practicality, this study has several important implications. First, tracking sensory satiety variability in aesthetic products can boost consumers' intent to purchase and consume. Satiation presents a challenge for prolonged sensory pleasure [29]. In a product's life cycle, the consumer product replacement cycle is also relatively short. Consumers who expect sensory pleasure and feel satiated by aesthetic products tend to buy new products. Of course, from a positive viewpoint, this can revitalize the market economy, but excessive consumption of unnecessary products can lead to waste. Through this study, we realized that one of the ways to encourage smart consumption habits in consumers is to emphasize sensory satiety's reduction effect.

Second, the findings suggest long-term strategies to enhance product loyalty for marketing managers through the management of aesthetic products. Among them, many companies preferred the brand-accessories strategy as a means to reduce consumer satisfaction. It is a strategy that continuously provides accessory products to the focal product to reduce the feeling of fullness and continuously provide new sensory stimulation [12,45]. For example, Mattel Inc. carried out a practical brand-accessories strategy that involved developing accessories for the product Barbie, such as clothes, a home, a car, and Barbie's boyfriend, Ken, even further. Therefore, Barbie customers believe that they can continue to enjoy the brand, and do not feel satiety. However, these accessory strategies can be costly, as they must be continually developed to meet consumers' aesthetic desires. Based on the findings of this study, marketers can take advantage of the dynamic effect of sensory satiety over time by planning and implementing incentive promotions for consumers who need to break their satiation.

## 5.3. Limitations and Future Research

This study demonstrates the effect of sensory satiety on perceived benefits and the differences in both dependent and independent variables over time. There are some limitations that suggest caution in assessing our findings. First, it provides various experiences of products to explain how aesthetic product sensory satiety might be delayed. In this study, the results showed that sensory satiety and perceived benefits decrease over time. According to existing research, amnesia of sensory satiety may be caused by interference with new stimuli and experiences. Galak et al. [29] stated that recalling a variety of product experiences may reduce sensory satiety. It is expected that repeated exposure to aesthetic products can lead to sensory satiety, which accelerates amnesia.

Second, it should be noted that the difference in sensory satiety is affected by amnesia depending on the individual's working memory ability—in particular, the consumer's working memory process. This means that there is a difference in individual working memory ability and a difference in the speed at which consumers experience sensory satiety from aesthetic products [16]. For instance, consumers with a high working memory ability would immediately experience sensory satiety from aesthetic products compared to those with a low working memory ability. It can be expected that the amnesia of sensory satiety is fast as well as constant with external stimuli. Therefore, in order to explain the effect of individual characteristics on memory amnesia more thoroughly, future studies should consider other potential determinants, as well as individual ability.

Third, we aimed to verify whether the difference in sensory satiety is affected by the sensory stimulation level based on the level of individual optimal stimulation. It is foreseeable that enhancing sensory stimulation level, which is a determinant of the optimal stimulation level, might induce varying behavioral responses. For instance, sensory satiety is diluted, and the consumer retains the aesthetic product for a relatively long time when igniting an individual's sensory capacity. Optimal

stimulation levels can be an informative factor for exploring the relationship between sensory satiety and amnesia far more deeply.

Fourth, there were not enough participants in the online survey. Panel members agree to carefully complete online surveys, and panel companies have diligence and accuracy checks as they respond to questions. However, statistically, the survey has garnered a sample that may not accurately represent the population. Some people are reticent to join panels, and so some members of the general population are not included. Future studies should consider other survey data collection methods.

Further research will be extended across different contexts on the sensory satiety effect by recalling the consumption experience of various products; it will be able to refine our knowledge on the influence of sensory satiety on consumers' behaviors. We would like to conduct a comparative study by classifying each product category with a level of either high or low involvement. Ultimately, there are ample opportunities for further research in the field of sensory satiety. Many factors remain unknown, in addition to reasons and mechanisms that can explain why dynamic effects between satiety and other sensory modalities, emotions, and cognitive responses occur.

**Author Contributions:** J.-E.J. and E.M.L. conceived of the framework of this paper. J.-E.J. developed the research model, collected data, analyzed the data, and wrote the paper. E.M.L. developed the research objectives and designed the coordination mechanism. All authors have read and agreed to the published version of the manuscript.

**Funding:** This research received no external funding.

**Conflicts of Interest:** The authors declare no conflict of interest.

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
