# Peer review of "The Effect of Sensory Satiety on Perceived Benefits: The Case of Aesthetic Consumption in South Korea"

_sustainability, doi:10.3390/su12208637_

Round 1

Reviewer 1 Report

The paper in general terms is well written, and has a pertinent topic. 

A general concern is about to the results and its discussion with the literature. It should be more detailed and more well fundamented. 

The authors should consider the following recommendations in order to improve the original manuscript:

  • A much more suitable title would be “The Effect of Sensory Satiety on Perceived Benefits: The Case of Aesthetic Consumption in South Korea”.
  • The idea of sustainability needs to be more visible in this research paper.
  • To extend Literature Review section by providing more relevant literature review, especially studies conducted during the last 5 years. There is no logical order in the literature review. All citations are in one flow. Yet, sentences look like a cut from cited papers and placed in this section without explaining why.
  • To expand the managerial implications in the article.
  • The sources must be added under each table.
  • Deepen the description of the limitations of conducted research and indicate the trends for further empirical research.
  • I would like to see a well-developed discussion comparing and contrasting solution/results presented in the work with existing work and then a subsection of it presenting contributions to theory/knowledge/literature and followed by a subsection on Implications for practice.

Regarding consumer behaviour framework, I suggest extending the literature section by including at least the following relevant studies:

  1. Hawaldar, I.T.; Ullal, M.S.; Birau, F.R.; Spulbar, C.M. Trapping Fake Discounts as Drivers of Real Revenues and Their Impact on Consumer’s Behavior in India: A Case Study. Sustainability 2019.
  2. Bonera, M.; Corvi, E.; Codini, A.P.; Ma, R. Does Nationality Matter in Eco-Behaviour? Sustainability 2017, 9, 1694.
  3. Antonides, G. Sustainable Consumer Behaviour: A Collection of Empirical Studies. Sustainability 2017, 9, 1686.

Human proofreading, English grammar and spelling correction are also required in order to improve the quality of the manuscript.

Reviewer 2 Report

The manuscript entitled “The Effect of Sensory Satiety on Perceived Benefits: The Case of Aesthetic Consumption” presents an interesting issue associated with the aesthetic consumption.

ABSTRACT:

  • Scientific writing has traditionally been third person, passive voice – author should avoid first person.
  • Please add the results accompanied the p-Values.
  • The most important findings (results) should be indicated and emphasized.

MATERIALS AND METHODS:

  • More detailed information associated with the recruitment are required.
  • Was the questionnaire previously validated? What was the accuracy and consistency of this questionnaire?
  • The most important aspect of the methodology is the applied tool. How did authors developed the tool? On the basis on what assumptions? How did authors verified the accuracy of the tool? This information must be provided and the applied methodology must be justify.
  • This section is crucial for whole study therefore it must be presented in more detailed way.
  • For the research that involves human subjects the rules of the  Declaration of Helsinki of 1975 must be applied (even for on-line survey), including ethics commission approval and especially informed consent. Please add the information about number of ethics commission approval (specific reference)

RESULTS and DISCUSSION:

  • Table 2 – two decimal places are sufficient.
  • In table 2 for each period, there are only 93 (or 97) respondents. Such group of not enough for the online survey.
  • In table 3, the p-Value should be added. Please do not present the correlation between the same discriminant.
  • Lines 280, 299 (etc.) – please indicate p-Value instead “n.s.” or “p<0.05”
  • Table 4 must be improved.
  • There is no almost discussion, but authors rather repetition of the results.. This section must be totally rewritten. Authors should relate the findings to those of similar studies and point the differences and similarities between the studies. Authors should add the appropriate references in this section.
  • In limitation section, authors should add limitations associated with the applied tool and the small numbers of respondents.

CONCLUSIONS:

  • Authors should add this section to emphasize the findings.

Other comments:

  • Authors should follow the Instructions for authors while preparing their manuscript (please see references).

Reviewer 3 Report

Introduction

The research problem and research questions have been well presented.

Literature Review

The author explains the research variables well. The discussion includes theory and previous research.

Research Method

The research method is well described. Research indicators are explained. Tests the goodness of data which includes reliability and validity and the analytical tools used are conveyed well. However, there are things that need the author's attention. Specifically, when participants were asked to consider the aesthetic products they already have. After that, each question regarding satiety, functional benefits, and emotional benefits of the product selected by the participant was measured. Thus, the research objects (namely the aesthetic products that the research respondents have) are diverse. How can the results of the respondent's answer from the respondents' thoughts on various products be averaged? The author also writes in the research methodology section as follows "The manipulation and measurement items of the main constructs used in this study are as follows". The question is: what is being manipulated?

Results and Discussion

Results and discussions are well written.

Conclusion

Research limitations and recommendations for further research are well presented.

References

More than 50 percent of the references in this manuscript are out of date (more than 10 years).

Round 2

Reviewer 1 Report

The authors have significantly improved the initial version of the manuscript. They incorporated all requested changes based on the previous review round.

Author Response

JOURNAL REVISION LETTER

October, 2020

Sustainability-954885

Title: The Effect of Sensory Satiety on Perceived Benefits: The Case of Aesthetic Consumption

Dear Reviewer,

We would like to thank you for the letter, and the opportunity to resubmit this revised manuscript. We would also like to take this opportunity to express our gratitude to the reviewers for their valuable and thoughtful comments on our earlier draft.

We believe that you have made several suggestions that would improve the overall quality of our manuscript.

The manuscript has been revised to address the reviewer comments, which are appended alongside our responses to this letter. We hope that you will agree with our belief that these revisions have greatly enhanced our manuscript.

The changes to the manuscript are indicated in red font.

Plesase see the attahment

Again, we appreciate the comments from the reviewers. If there are any additional concerns or comments, please do not hesitate to contact me.

Thank you for reviewing our manuscript.

Sincerely yours,

Authors

Response to Reviewer 1 Comments

Point 1:

The authors have significantly improved the initial version of the manuscript. They incorporated all requested changes based on the previous review round.

- We appreciate your helpful comments. Overall, we can improve this research. Again, we appreciate the comments from reviewer 1.

Reviewer 2 Report

I appreciate the great efforts that the authors have made in response to my questions and concerns. The revision clarifies almost all the points, but I have some additional comments :

  • Discussion section was not improved sufficiently (some important references are still missing). Some detailed information about other studies are necessary.
  • In table 3 - there is some missing number in “** p – Value = .000” (maybe p<.001?)
  • There are some problems with formatting (e.g. style in references section; style of tables)

Author Response

JOURNAL REVISION LETTER

October, 2020

Sustainability-954885

Title: The Effect of Sensory Satiety on Perceived Benefits: The Case of Aesthetic Consumption

Dear Reviewer,

We would like to thank you for the letter, and the opportunity to resubmit a revised copy of this manuscript. We would also like to take this opportunity to express our thanks to the reviewers for your valuable and thoughtful comments on our earlier draft.

We believe that you have made a number of suggestions that would improve the overall quality of our manuscript. The manuscript has been revised to address the reviewer comments, which are appended alongside our responses to this letter. We hope that you will agree with our belief that these revisions have greatly enhanced our manuscript.

We revised manuscript using the red words.

Again, we appreciate the comments from the reviewers. If there are any additional concerns or comments, please do not hesitate to contact me.

Thank you for reviewing our manuscript.

Sincerely yours,

Authors

Response to Reviewer 2 Comments

Point 1:

Discussion section was not improved sufficiently (some important references are still missing).

- We appreciate your helpful comments. Overall, we have revised the implication section.

(in the text) “Although many researchers equate satiation with reaching a physiological limit, some evidence suggests that satiation can also result from cognitive processes. If consumers’ sensory systems are repeatedly exposed to a product or brand’s aesthetic design, satiation from accustomedness will follow due to the activation process of various pieces of information related to the products. According to Galak et al. [29], sensory satiety can be reduced again because, as a cognitive response, consumers can lose information related to the stimulus. This is meaningful for empirically revealing that sensory satiety of aesthetic products may change over time.” (See line 384-391)

 (in the text) “In this study, sensory satiety dealt with accustomedness and satiation resulting from repeated use of aesthetic products. However, perceived benefits refer to subjective compensation in customers' expectations that the product will satisfy their needs [4,10,18]. Thus, our theory provides some insight into the effects of sensory satiety on perceived benefits; furthermore, it offers prescriptive suggestions. Breaking off from a repeated activity can reduce hedonic adaptation, but consumers do not appreciate this [44]. However, our focus is on the fact that the aesthetic design of products should be tailored to the consumers’ working memory ability. Additionally, sensory satiety can lead to dynamic changes (e.g., decreases in pleasure and interest) over time due to continuous use of a product, while the difference in perceived benefits is static regardless of time [16,19,29]. This research provides insights for understanding how consumers recover from aesthetic design satiation.(See line 393-403)”

(in the text) “In terms of practicality, this study has several important implications. First, tracking sensory satiety variability in aesthetic products can boost consumers' intent to purchase and consume. Satiation presents a challenge for prolonged sensory pleasure [29]. (See line 404-406)

Point 2:

In table 3 - there is some missing number in “** p – Value = .000” (maybe p<.001?)

- We appreciate your helpful comments. According your comments, we revised table 3

P – Value = .000 -> p <.001 (See line 290)

Point 3:

There are some problems with formatting (e.g. style in references section; style of tables)

- Thank you for your detailed comments. We have revised the manuscript following the journal guidelines.